# MatrixNet: Learning over symmetry groups using learned group representations

**Lucas Laird**
Khoury College of Computer Sciences
Northeastern University
Boston, MA 02115
`laird.l@northeastern.edu`

**Circe Hsu**
Department of Mathematics
Northeastern University
Boston, MA 02115
`hsu.circe@northeastern.edu`

**Asilata Bapat**
Mathematical Sciences Institute
Australian National University
Canberra, Australia
`asilata.bapat@anu.edu.au`

**Robin Walters**
Khoury College of Computer Sciences
Northeastern University
Boston, MA 02115
`r.walters@northeastern.edu`

## Abstract

Group theory has been used in machine learning to provide a theoretically grounded approach for incorporating known symmetry transformations in tasks from robotics to protein modeling. In these applications, equivariant neural networks use known symmetry groups with predefined representations to learn over geometric input data. We propose MatrixNet, a neural network architecture that learns matrix representations of group element inputs instead of using predefined representations. MatrixNet achieves higher sample efficiency and generalization over several standard baselines in prediction tasks over the several finite groups and the Artin braid group. We also show that MatrixNet respects group relations allowing generalization to group elements of greater word length than in the training set. Our code is available at https://github.com/lucas-laird/MatrixNet.

## 1 Introduction

The choice of representation for input features is a key design aspect for deep learning. While a wide variety of data types are considered in learning tasks, for example, sequences, sets, images, time series, and graphs, neural network architectures admit only tensors as input. Various methods exist for mapping different input types to tensors such as one-hot embeddings, discretization of continuous signals, learned token embeddings of image patches or words [1, 2], adjacency matrices [3], positional encodings [1], or spectral embeddings [4].

In this paper we consider the question of what feature representations to use for learning tasks with inputs coming from a symmetry group. There are many examples of tasks defined over symmetry groups, such as policy learning in robotics [5], reinforcement learning [6], pose estimation in computer vision [7], sampling states of quantum systems [8], inference over orderings of a set [9], and group-theoretic invariants in pure mathematics. Past work has typically employed fixed representations chosen from among the known representation theory of the group. Representation theory is the branch of mathematics concerned with classifying the set of representations of a group $G$ which, in this context, refers to homomorphic realizations of a group in terms of $n \times n$ matrices. For groups with well understood representation theory, for example, the symmetric group $S_n$ or $SO(n)$, this provides a ready set of embeddings for converting group elements into tensors for use in downstream

38th Conference on Neural Information Processing Systems (NeurIPS 2024).

models. Trial and error or topological analysis has shown that the choice of group representation is critical for learning [10, 11, 12, 13].[1]

Instead of using predefined group element representations, we propose to learn feature representations for each group element using a learned group representation. That is, we learn to map group elements to invertible $n \times n$ matrices which respect the symmetry group structure. There are several advantages to this strategy. First, unlike predefined group representations, learned representations allow the model to adapt to the given task and capture relevant information for the learning task. Second, learned representations provide reasonable correlations between the learned features for different group elements since they incorporate algebraic structure into the model. This structure is encouraged using free group generators and group relation regularization. Third, relative to learned vector embeddings, learned matrix representations are very parameter efficient for encoding different group elements, reducing the problem to learning only representations of group generators; using sequence encoding, our method is able to generalize to combinatorially large or even infinite groups. Fourth, the learned representation admits analysis in terms of the irreducible subspaces of the generators, giving insight into the model's understanding of the task.

We integrate our learned group representation into a specialized architecture, MatrixNet, adapted for learning mappings related to open problems in representation theory. We compare against several more general baselines on order prediction over finite groups and estimating sizes of categorical objects under an action of the Artin braid group. Through our experiments we observe that our approach achieves higher sample efficiency and performance than the baselines. We additionally show that MatrixNet's constraints allow for it to generalize to unseen group elements automatically without the need for additional data augmentation.

Our contributions include:

- Formulation of the problem of learning over groups as a sequence learning problem in terms of group generators and invariance to group axioms and relations,
- The MatrixNet architecture, which utilizes learned group representations for flexible and efficient feature learning over symmetry groups,
- The matrix block method for constraining the network to respect the group axioms and an additional loss term for learning group relations,
- Empirical validation showing MatrixNet outperforms baseline models on a task over the symmetric group and a task over the braid group related to open problems in mathematics.

## 2  Related Works

**Mathematically Constrained Networks**   Many deep learning methods incorporate mathematical constraints to better model underlying data. For instance, the application of graph neural networks [3, 14] to problems with an underlying graph structure has led to state of the art performance in many domains. Deep learning models have also been designed for tasks with known mathematical formulations by parameterizing components of algorithmic solutions as neural networks and leveraging their structures for more efficient optimization [15, 16]. More broadly the field of geometric deep learning [17] advocates for building neural networks which reflect the geometric structure of data in their latent features. When symmetries are present in data, group equivariant neural networks [18, 19, 20, 21] can enable improved generalization and data efficiency by incorporating known symmetries into the model architecture using the representation theory of groups. Our method also utilizes group representations, but unlike equivariant neural networks, we use learned as opposed to fixed representations. We also focus on modeling functions defined on the group as opposed to between representations of the group.

**Learning Structured Representations**   Instead of using predefined representations as inputs, many methods seek to learn mathematically structured representations from data. This idea has been applied in physics [22, 23], robotics [24], world models [25], self-supervised learning [26, 27], and unsupervised disentanglement [28]. Park et al. [29], for example, use a combination of

---

[1]The term representation is used in both deep learning and mathematics with related but different meanings. We will disambiguate the former as a *feature representation* and the latter as a *group representation*.

learned symmetry and equivariant constraints to map images to group elements or vectors in group representations. Similar techniques are used in symmetry discovery where the underlying group symmetry is not known a priori [30, 31, 32]. Yang et al. [32] use a generative model to learn latent representations with a linear group action in order to find unknown group symmetries in data. Here, in contrast, we start with a known group and learn a matrix representation. Hajij et al. [33] propose algebraically-informed neural networks which learn a non-linear group action from a group presentation. We also learn a group action but consider linear group actions and the learning signal comes not only from the group presentation but also a downstream task.

**Deep Learning for Math**   Recent work has shown deep learning can be useful for providing examples, insight, and proofs related to open problems in mathematics. One approach is the application of language models to mathematics [34], which has the benefit of flexibility in how the model is prompted yet is difficult to interpret and prone to errors. Meanwhile significant work has been done in the area of symbolic regression and automated theorem proving [35]. Other work applies deep learning to the direct modeling of partial differential equations [36, 37]. These methods can perform exceptionally well on real-world data [38], but suffer when trying to interpret predictions made for the purposes of mathematical research. Another avenue involves training graph neural networks on mathematical data such as group or knot invariants and analyzing the learned representations to see which features are significant as a way to provide intuition to mathematicians [39] . Our method also uses structured inputs and learned features, but uses a sequence model and learned group representation instead of a graph neural network with learned node attributes.

## 3   Background

### 3.1   Group Theory

Groups encode systems of symmetry and have been used in machine learning to build invariance into neural networks to various transformations [18]. Formally, a **group** $G$ is a set equipped with an associative binary operation $\circ \colon G \times G \to G$ which satisfies two axioms: (1) there exists an identity element $1 \in G$, such that $g \circ 1 = 1 \circ g = g$, (2) for each $g \in G$, there exists an inverse $g^{-1} \in G$ such that $g \circ g^{-1} = g^{-1} \circ g = 1$. Examples of groups include finite groups such as the dihedral group $D_4$ which gives the symmetries of the square, $\mathrm{SO}(3)$, the continuous group of 3D rotations, or $(\mathbb{Z}, +)$ the infinite discrete group of integer shifts.

Since groups may be combinatorially large or infinite, it is essential to encode their elements and compositional structure in a succinct way. For many discrete groups, generators and relations provide such a description. A set of elements $S = \{g_1, ..., g_n\} \subseteq G$ are called **generators** if every element of $G$ can be written as a composition of $g_1, g_1^{-1}, ..., g_n, g_n^{-1}$. In general, each element of $G$ may be written in many different ways in terms of the generators; this non-uniqueness is encoded using a set of relations. A set $R = \{r_1, \ldots, r_m\}$ of words in the generators $S$ are **relations** for $G$ if each word $r_i$ is equal to the identity in $G$ and if $R$ generates the entire set of words equal to identity under composition and conjugation. The generators and relations of a group taken together are called a **presentation** and denoted $G = \langle g_1, ..., g_n \mid r_1, ..., r_m \rangle$. For example $D_4 = \langle r, f \mid r^4 = f^2 = frfr \rangle$. Due to relations, group elements do not have a unique word representation. For example $frf = r^3 = r^7$ all represent the same group element. By convention relations are sometimes stated as equalities instead of single group elements, for example $frf = r^3$. The **free group** $F_S = \langle g_1, \ldots, g_n \rangle$ is defined to have no relations except for those coming from the two group axioms above.

An important notion for our discussion is the **order** of an element. If $g \in G$ then the order of $g$, denoted $|g|$, is the smallest $k$ such that $g^k = e$. (For non-finite groups, $k$ may be infinity). The order of the group $|G|$ is simply the number of elements in the group. For any $g$, Lagrange's theorem implies that $|g|$ is a divisor of $|G|$ [40], which restricts the possible orders an element may take.

### 3.2   Representation Theory

Abstract group presentations are difficult to work with in many settings. Group representations map group elements to invertible matrices such that composition of group elements corresponds to matrix multiplication. This gives the group a natural action on vector spaces and allow for analysis of the group using linear algebra. Formally, a **representation of a group** $G$ is a group homomorphism

$\Phi : G \to \mathrm{GL}(n)$ to the group of invertible $n \times n$ matrices. That is $\Phi(g_1 \cdot g_2) = \Phi(g_1) \cdot \Phi(g_2)$. A property of $\Phi$ is that $\Phi(1) = I_{n \times n}$ and $\Phi(g^{-1}) = \Phi(g)^{-1}$. Due to the homomorphism property, it is sufficient to define $\Phi$ for generators of the group $G$. For example, a $2 \times 2$ representation of $D_4$ is given by mapping $r$ to a $\pi/2$-rotation matrix and $f$ to a reflection over the $x$-axis.

The representations of many groups are well classified. This provides a ready source of tensor representations for group elements to use as inputs for neural networks. For example, for finite groups, by Maschke's Theorem [41], representations may be decomposed into **irreducible representations**. That is, there exists a basis such that the representation matrices are all block diagonal with the same block sizes and these blocks cannot be further subdivided. The irreducible representations may then be further classified by computing character tables.

### 3.3 Symmetric and Braid Group

The **braid group** on $n$ strands $B_n$ has presentation

$$\langle \sigma_1, ..., \sigma_{n-1} \mid \sigma_i \sigma_j = \sigma_j \sigma_i \text{ for } |i - j| \geq 2 \, , \sigma_i \sigma_{i+1} \sigma_i = \sigma_{i+1} \sigma_i \sigma_{i+1} \text{ for } 1 \leq i \leq n - 2 \rangle. \quad (1)$$

The braid group intuitively represents all possible ways to braid a set of $n$ strands. The generators $\sigma_i$ correspond to twisting strand $i$ over $i + 1$ and $\sigma_i^{-1}$ is the reverse, twisting strand $i + 1$ over $i$. It is defined topologically as equivalence classes up to ambient isotopy of $n$ non-intersecting curves in $\mathbb{R}^3$ connecting two sets of $n$ fixed points. The braid group is infinite and though some representations are known, they are not fully classified [42]. The braid group has important connections to knot theory, mathematical physics, representation theory, and category theory.

The **symmetric group** on $n$ elements, denoted $S_n$, is defined as the set of bijections from $\{1, \ldots, n\}$ to itself. It is also a quotient of the braid group $B_n$ and has a presentation similar to Eqn. 1 but with additional relations $\sigma_i^2 = 1$ for $1 \leq i \leq n - 1$. Here $\sigma_i$ is the transposition $(i \ \ i+1)$. The symmetric group has finite order $|S_n| = n!$. Representations of the symmetric group are well understood. The irreducible representations are parameterized by partitions of $n$. For more details, see [43].

### 3.4 Categorical Braid Actions

One current active research problem in mathematics concerns actions of braid groups on categories. A category is an abstract mathematical structure that has *objects* and maps or *morphisms* between objects, satisfying several coherence axioms. For example, the category $\mathrm{Vect}_{\mathbb{R}}$ has objects which are real vector spaces and morphisms which are linear maps. *Functors* are maps between categories that take objects to objects and morphisms to morphisms between the corresponding objects, satisfying several compatibility conditions. The action of a group $G$ on a category $\mathcal{C}$ means that each group element $g \in G$ is associated with an invertible functor $F_g : \mathcal{C} \to \mathcal{C}$, such that any relation $x = y$ in the group implies that the corresponding functors $F_x$ and $F_y$ are naturally isomorphic.

Given a category on which a braid group acts, mathematicians are interested in measuring how objects grow under repeated applications of elements of the braid group. The "size" of an object in a category may be measured using a tool called Jordan–Hölder filtrations. For example, Bapat et al. [44] attempt to measure growth rates of objects in a specific category $\mathcal{C}_n$ under repeated applications of certain twist functors $\sigma_{P_i}$, which define an action of the braid group $B_n$ on $\mathcal{C}_n$. Each object in the category has a Jordan–Hölder filtration giving a unique vector in $\mathbb{Z}_{\geq 0}^n$ of Jordan–Hölder multiplicities. For more details see [44] and Appendix A.

However, they are only able to compute the action in certain cases and a simple formula is elusive. A complete description of the Jordan–Hölder multiplicities after applying combinations of $\sigma_{P_i}$ to one of the generating objects is only known for $n = 3$; that is, the case of the 3-strand braid group $B_3$. Understanding how these multiplicities evolve under repeated application of braids is a challenging open research problem in mathematics.

## 4 Methods

We formulate the problem of learning a function on a symmetry group as a sequence learning problem using a presentation of the group in terms of generators and relations. We propose MatrixNet which predicts the label using a learned matrix representation for the group. The homomorphism property of the representation is enforced through a combination of model design and an auxiliary loss term.

### 4.1 Problem Formulation

We consider task functions of the form $f: G \to \mathbb{R}^c$ where $G$ is a finite or discrete group and the output space $\mathbb{R}^c$ may represent either a regression target or class label. While such tasks appear in computer vision, robotics, and protein modeling, we are particularly interested in problems in mathematics where neural models may lend additional examples and insight towards proving theorems.

To efficiently represent group elements in infinite or large groups, we consider a presentation of $G = \langle S \mid R \rangle$ in terms of generators $S = \{g_1, \dots, g_n\}$ and relations $R = \{r_1, \dots, r_m\}$. Model inputs $g \in G$ are represented by sequences of generators $(g_{i_1}, \dots, g_{i_\ell})$ where $g = g_{i_1} \circ \dots \circ g_{i_\ell}$ is of arbitrary length $\ell \geq 0$ and $1 \leq i_j \leq n$. For convenience, we can include the identity $g_0 = e$ among the generators to pad sequences without changing the group element.

Since a single group element may be represented by different sequences, it is critical for the model $f_\theta$ to be invariant to both the group axioms and relations. That is, we desire

$$f_\theta(g_{i_1}, \dots, g_{i_k}, e, g_{i_{k+1}}, \dots, g_{i_\ell}) = f_\theta(g_{i_1}, \dots, g_{i_k}, g_{i_{k+1}}, \dots, g_{i_\ell}) \quad \text{(Identity)}, \quad (2)$$

$$f_\theta(g_{i_1}, \dots, g_{i_k}, g_j, g_j^{-1}, g_{i_{k+1}}, \dots, g_{i_\ell}) = f_\theta(g_{i_1}, \dots, g_{i_k}, g_{i_{k+1}}, \dots, g_{i_\ell}) \quad \text{(Inverses)}, \quad (3)$$

$$f_\theta(g_{i_1}, \dots, g_{i_k}, r_j, g_{i_{k+1}}, \dots, g_{i_\ell}) = f_\theta(g_{i_1}, \dots, g_{i_k}, g_{i_{k+1}}, \dots, g_{i_\ell}) \quad \text{(Relations)}. \quad (4)$$

### 4.2 MatrixNet

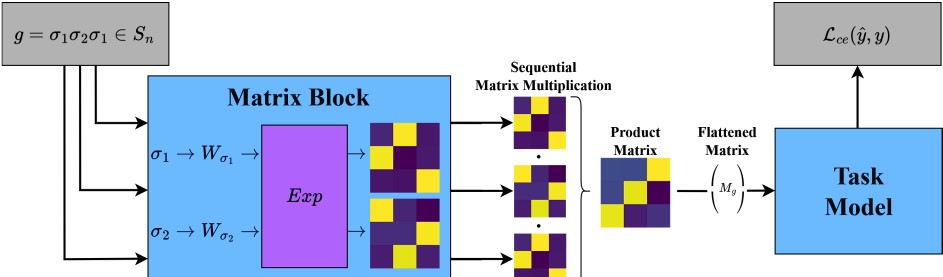

Figure 1: Schematic of MatrixNet for predicting order of elements of $S_3$. Input generators $\sigma_1$ and $\sigma_2$ are mapped to learned representations and sequentially multiplied to provide a matrix representation of group element $g$. The order is then predicted by the task model which is an MLP.

We propose MatrixNet (see Figure 1), a neural sequence model which models functions on a group $G$ by taking as input sequences of generators for $G$. It achieves invariance to group axioms and relations through a combination of built in constraints and an auxiliary loss term. The key part of the MatrixNet architecture is the matrix block which takes a group generator $g_i$ as input and outputs an invertible square matrix representation $M_{g_i}$. The matrix representation $M_g$ for an arbitrary group element $g$ is defined as the product of matrix representations of generators needed to generate $g$. This matrix representation is then flattened and passed to a downstream task model (such as an MLP) which computes the label. In what follows, we give a more detailed description of the matrix block and some variations on the architecture.

**Signed one-hot encoding** We define the *signed one-hot encoding*, a modified version of the traditional one-hot encoding, for encoding group generators used as input to MatrixNet. Let $(g_{i_1}^{\epsilon_1}, g_{i_2}^{\epsilon_2}, \dots, g_{i_\ell}^{\epsilon_\ell})$ be a sequence of generators where $0 \leq i_k \leq n$ and $\epsilon_k \in \{\pm 1\}$. The signed one-hot encoding encodes each generator $g_{i_k}^{\epsilon_k}$ as a vector $v_{g_{i_k}} = \epsilon e_i = [0, \dots, 0, \epsilon_k, 0, \dots, 0] \in \mathbb{R}^n$ which is 1 or -1 in the $i$th entry. The identity element $g_0 = 1$ is mapped to the zero vector $v_1 = \mathbf{0} \in \mathbb{R}^n$. The signed one-hot encoding is chosen since it intuitively relates a generator and its inverse as $v_{g^{-1}} = -v_g$.

#### 4.2.1 Matrix Blocks and Learned Matrix Representations

The matrix block is designed as a parameterized representation of the free group $F_S$, that is, a homomorphism $\Phi : F_S \to GL(n)$ which satisfies 3 properties: (1) the matrix $\Phi(g) = M_g$ is invertible, (2) $\Phi(1) = I_{n \times n}$, and (3) if $\Phi(g) = M_g$ then $\Phi(g^{-1}) = M_g^{-1}$. In what follows $v_{g_{i_k}}$ is

the signed one-hot encoding of the generator $g_{i_k}$ and $W$ is a learnable parameter matrix. For a group element $g = g_{i_1} \ldots g_{i_n}$, the matrix block is defined

$$A_k = \mathrm{Reshape}(W v_{g_{i_k}})$$
$$M_{g_{i_k}} = \mathrm{MatrixExp}(A_k)$$
$$M_g = M_{g_{i_1}} M_{g_{i_2}} \ldots M_{g_{i_n}}$$

where Reshape reshapes a vector in $\mathbb{R}^{n^2}$ into a square $n \times n$ matrix.

**Proposition 1.** *Matrix Block defines a representation of the free group.*

*Proof.* Property (1) is satisfied since the outputs of a matrix exponential are invertible. Properties (2) and (3) follow from $v_{g_{i_k}^{-1}} = -v_{g_{i_k}}$, $v_1 = \mathbf{0}$, and properties of the matrix exponential,

$$M_1 = \mathrm{MatrixExp}(\mathbf{0}_{n \times n}) = I_{n \times n}$$
$$M_{g_{i_k}^{-1}} = \mathrm{MatrixExp}(-A_k) = M_{g_{i_k}^{-1}}.$$

$\square$

### 4.2.2 Variations of Matrix Block

We now present some variations of this simple design that satisfy the group homomorphism properties.

**Linear Network (MatrixNet-LN)** The first variant replaces the single parameter matrix $W$ with a linear network that has two parameter matrices $W_1, W_2$. The linear network matrix block changes the computation of the intermediate $A_k$ matrix to:

$$A_k = \mathrm{Reshape}(W_2 W_1 v_{g_{i_k}})$$

This is still a linear function of $v_{g_{i_k}}$ meaning Proposition 1 still holds by the same argument.

While this variation does not give increased expressivity over the original formulation of the matrix block, the linear network can change the optimization landscape leading to different performance in practice. This variation is called MatrixNet-LN.

**Non-Linearity (MatrixNet-NL)** The second variation introduces an element-wise odd non-linear function $f$. That is $f(-x) = -f(x)$ as with $\tanh$. The non-linear matrix block modifies

$$A_k = \mathrm{Reshape}(f(W_1 v_{g_{i_k}}))$$

**Proposition 2.** *Non-linear matrix block defines a representation of the free group.*

*Proof.* Property (1) is satisfied by the same argument in Proposition 1. Since $f$ is an odd function, $f(W_1 v_1) = f(\mathbf{0})$ and $\mathrm{Reshape}(f(W_1 v_{g_{i_k}^{-1}})) = \mathrm{Reshape}(f(-W_1 v_{g_{i_k}})) = -A_k$. Therefore Properties (2) and (3) are satisfied by the same argument in Proposition 1. $\square$

This variation can be combined with the first as Proposition 2 holds for linear transformations of $A_k$. That is, $A_k = \mathrm{Reshape}(W_2 f(W_1 v_{g_{i_k}}))$. Unless otherwise noted we set $f = \tanh$.

**Block-Diagonal (MatrixNet-MC)** The third variation is inspired by the decomposition of representations into irreducible representations. For certain classes of groups such as finite groups, every representation decomposes such that the matrices $M_g$ have a consistent block diagonal structure in some basis. Thus to learn an arbitrary representation of the group $G$, it suffices to learn a block diagonal representation assuming the blocks are large enough.

This variation learns $\ell$ intermediate $n_j \times n_j$ matrices $A_{k_j}$ which are combined to form a block-diagonal $n \times n$ matrix $A_k$ where $n = \sum_{j=1}^{\ell} n_j$. The block-diagonal matrix block is defined with

$$A_{k_j} = \mathrm{Reshape}(W_j v_{g_{i_k}}) \text{ for } j = 1 \text{ to } \ell$$
$$A_k = \mathrm{BlockDiag}(A_{k_1}, A_{k_2}, \ldots, A_{k_\ell})$$

Note that $\mathrm{MatrixExp}$ and matrix multiplication preserve the block structure. If the sizes $n_j$ are fixed to all be equal, this formulation can be implemented as a multi-channel matrix block where both $A_k$ and $M_{g_{i_k}}$ are $\ell \times n_j \times n_j$ tensors with $\ell$ channels. Each $A_{k_j}$ is calculated identically to $A_k$ in the original matrix block formulation, and $\mathrm{BlockDiag}$ is linear, so Proposition 1 still holds. This variation is also compatible with the previous two variations. In our experiments, we implement a 3-block version called MatrixNet-MC with a single linear layer and no non-linearity.

### 4.3 Enforcing group relation invariances

The matrix block is constrained to learn a representation of the free group $F_S$. As a consequence MatrixNet will satisfy (2) and (3) as desired. However, most groups have relations which cannot be enforced through a simple weight-sharing scheme used in equivariant architectures [19]. We propose to learn the relations through a secondary loss which measures how closely the representation respects the group relations. More concretely, let $G = \langle S|R \rangle$ be a group with relations $r_i \in R$. The loss is:

$$\mathcal{L}_{rel} = \sum_{r_i \in R} (||M_{r_i} - I_{n \times n}||)$$

Since MatrixNet learns a representation that is invariant to group axioms, it is sufficient to sum over only $\{r_i\}_{i=1}^m$ and not all compositions of relations. For architectures which do not respect the free group structure, the relations $r_i$ alone may not guarantee that all equivalent words have identical feature representations, requiring potentially combinatorial amounts of data augmentation. This allows MatrixNet to both efficiently learn group relation invariance and simply verify this invariance without any data augmentation.

## 5 Experiments

We use two learning tasks to evaluate the four variants of MatrixNet and compare our approach against several baseline models. We use several finite groups on a well understood task as an initial test to validate our approach and then move on to an infinite group, the braid group $B_3$, on a task related to open problems. As baselines, we compare to an MLP for fixed maximum sequence length and LSTM and Transformer models on longer sequences. Baseline model parameters were chosen so all of the models have approximately the same number of trainable parameters.

### 5.1 Order Prediction in Finite Groups

The first task is to predict the order of group elements in finite groups. Elements of finite groups are input as finite sequences of generators as described in Section 3.3. The typical efficient algorithm for computing the order involves disjoint cycle decomposition, making order classification a non-trivial task. See Appendix B.1 for more details on the sampling method and data splits.

**Models Compared** We compare MatrixNet variants and three baselines for order prediction in $S_{10}$: (1) **MLP** with 3 layers with hidden dimension 256 and SiLU activations, (2) **Fixed representation** input to a 2-layer classifier MLP with 256 hidden dimensions and SiLU activations, (3) **LSTM** input to a 2-layer LSTM with 256 hidden dimensions with a subsequent MLP classifier using SiLU activations, (4) **MatrixNet-LN** with a 2-layer 256 hidden dimension matrix block and classifier network with SiLU activations. (5) **MatrixNet-MC** with a $2x2$ matrix block size over 5 matrix channels and classifier network with SiLU activations. (6) **MatrixNet-NL** with a 2-layer 256 hidden dimension matrix block with SiLU activations and classifier network with SiLU activations. The precomputed representation is an ablated version of MatrixNet using a fixed representation of $S_{10}$ instead of a learned one. For the fixed representation, we use the standard $10 \times 10$ representation given by the permutation matrices corresponding to the group element. In MatrixNet-LN, the activation between layers of the matrix block is set to a linear passthrough while in MatrixNet-NL the activation is specified to be SiLU. MatrixNet-MC enforces a $2x2$ block diagonal structure on the learned representations corresponding to the 2-dimensional irreps of $S_{10}$.

We also note the use of SiLU activation in our $S_{10}$ MatrixNet model. Due to the generator self-inverse property we need not consider separate generator inverses, and so the odd function requirement given in Proposition 2 is not applicable.

Table 1: MatrixNet and baseline performance on $S_{10}$ order prediction

| Model | Parameters | CE Loss ($10^{-2}$) | Acc |
|---|---|---|---|
| MLP | 365584 | $0.02 \pm 0.01$ | $100 \pm 0$ |
| Rep Ablation | 299792 | $4.8 \pm 0.5$ | $87 \pm 2$ |
| LSTM | 270505 | $8.1 \pm 1.4$ | $77.3 \pm 5.2$ |
| MatrixNet-LN | 343968 | $0.9 \pm 0.3$ | $99.4 \pm 0.4$ |
| MatrixNet-MC | 82960 | $0.02 \pm 0.01$ | $100 \pm 0$ |
| MatrixNet-Nonlinear | 343968 | $0.0003 \pm 0$ | $100 \pm 0$ |

Table 2: MatrixNet performance on finite group order prediction

| Group | |G| | Rep. Size | Classes | CE Loss ($10^{-2}$) | Acc (%) |
|---|---|---|---|---|---|
| $S_{10}$ | $10^6$ | 10 | 16 | $0.0003 \pm 0$ | $100 \pm 0$ |
| $S_{12}$ | $10^8$ | 12 | 23 | $0.8 \pm 0.2$ | $99.2 \pm 0.4$ |
| $C_{11} \times C_{12} \times ... \times C_{15}$ | $10^5$ | 10 | 35 | $2.1 \pm 2.4$ | $87.3 \pm 18$ |
| $S_5 \times S_5 \times S_5 \times S_5$ | $10^8$ | 20 | 12 | $2.6 \pm 0.4$ | $98.3 \pm 0.3$ |

**Model Comparison Results**   Results of the experiments are summarized in Table 1. All variants of MatrixNet achieve a classification accuracy of at least 99% across multiple independent trials. Of note is the inferior performance of the precomputed representation baseline compared to the MLP and MatrixNet on both loss and accuracy metrics, suggesting that there is an advantage to a learnable representation. These results on $S_{10}$ order classification validate that group representation learning can aid learning of tasks defined over groups.

**Order Prediction over Different Groups**   In order to demonstrate the flexibility of MatrixNet, we show that MatrixNet can be used to predict order across several different sizes and types of groups. In addition to $S_{10}$, we evaluate MatrixNet on a larger symmetric group $S_{12}$ an Abelian group $C_{11} \times C_{12} \times C_{13} \times C_{14} \times C_{15}$ and a product $S_5 \times S_5 \times S_5 \times S_5$. These product groups provide a more complex group structure which MatrixNet must learn for successful generalization, with varying representation structure. The results for these experiments are summarized in Table 2.

MatrixNet achieves a high classification accuracy across all additional groups tested. However, accuracy for the Abelian group is lower than the accuracies for other groups tested (87% vs 99%). One explanation for this decrease in accuracy is due to the large number of valid orders of the group. Additionally, due to the structure of finite Abelian groups, many element orders will be underrepresented in random sampling.

## 5.2   Categorical Braid Action Prediction

In our second experiment, we train models to predict the Jordan–Hölder multiplicities from braid words in the braid group $B_3$ (see Section 3.4). The task is formulated as a regression task with a mean-squared error (MSE) loss function. The Jordan–Hölder multiplicities are integers, so we evaluate accuracy by rounding the vector entries to the nearest integer. This accuracy is reported as an average accuracy over the three entries of the Jordan–Hölder multiplicities vector. Elements of $B_3$ are generated by two generators $\sigma_1, \sigma_2$ and their inverses and are encoded using a signed one-hot encoding. For more details on the dataset generation process and data splits see Appendix B.2.

We additionally performed an experiment to evaluate how well MatrixNet generalizes to unseen braid words longer than those seen in training. For this experiment, we compare against the MLP and LSTM since these were the highest performing baselines.

**Baseline Comparison Results**   We trained all of the models for 100 epochs as all of the models except the Transformer converged within 100 epochs. Despite performance converging much faster than 100 epochs for most MatrixNet variants, we found that additional epochs of training improved the model's invariance to group relations with minimal variations in performance. Table 3 shows the performance of the baseline models and MatrixNet variations at 50 and 100 epochs of training

averaged over 5 runs. The simple MatrixNet model was the worst performing MatrixNet variant slightly outperforming the baseline models at 100 epochs. All other variants of MatrixNet vastly outperform baselines with both MatrixNet-LN and MatrixNet-NL achieving MSE far below the baselines and perfect or near perfect accuracy across all runs. These results confirm the results from the order prediction experiments and demonstrate the advantage of MatrixNet for learning over groups.

Table 3: MSE and accuracy of Jordan–Hölder multiplicities for baseline models and MatrixNet variations. Results are averaged over 5 runs. See Appendix B.2 for model parameters and training details.

| Model Type | Parameters | MSE Epoch 50 | MSE Epoch 100 | Avg. Acc. |
|---|---|---|---|---|
| Transformer | 63779 | $3.013 \pm 0.147$ | $2.895 \pm 0.024$ | $42.3\% \pm 1.2\%$ |
| MLP | 52099 | $0.315 \pm 0.004$ | $0.132 \pm 0.009$ | $89.1\% \pm 0.73\%$ |
| LSTM | 51027 | $0.345 \pm 0.149$ | $0.075 \pm 0.035$ | $93.0\% \pm 3.9\%$ |
| MatrixNet | 42507 | $0.543 \pm 0.458$ | $0.082 \pm 0.034$ | $95.1\% \pm 0.9\%$ |
| MatrixNet-LN | 42883 | $\mathbf{7.1e{-}4 \pm 2.4e{-}4}$ | $0.001 \pm 0.001$ | $\mathbf{99.9\% \pm 0.004\%}$ |
| MatrixNet-MC | 41987 | $0.063 \pm 0.033$ | $0.014 \pm 0.006$ | $98.8\% \pm 0.6\%$ |
| MatrixNet-NL | 42883 | $0.002 \pm 0.003$ | $\mathbf{6.4e{-}4 \pm 3.6e{-}4}$ | $\mathbf{99.9\% \pm 0.008\%}$ |

**Length Extrapolation Results**  The results in Figure 2 show how the MSE and average accuracy change as input length increases averaged over 10 runs. A single run was omitted from the results for MatrixNet due to training instability. We observe explosive MSE growth for MatrixNet and MatrixNet-MC, but both maintain higher average accuracy than the baselines. The high variance in MSE suggests that both variants are capable of extrapolating despite struggling compared to the other two variants. MatrixNet-LN and MatrixNet-NL both maintain near-zero average MSE as length increases and consequently achieve near perfect average accuracy as length increases.

The discrepancy in extrapolation performance of the MatrixNet variations suggest that MatrixNet-LN and MatrixNet-NL learn better representations than MatrixNet and MatrixNet-MC. To measure this, we compare the relational error of the four MatrixNet variations in Table 4. The group $B_3$ has only the braid relation $\sigma_1\sigma_2\sigma_1 = \sigma_2\sigma_1\sigma_2$. We calculate the relational error as the Fröbenius norm of the difference $||M_{\sigma_1}M_{\sigma_2}M_{\sigma_1} = M_{\sigma_2}M_{\sigma_1}M_{\sigma_2}||$. We also compute this distance between two non-equivalent braids $\sigma_1\sigma_1\sigma_2, \sigma_2\sigma_2\sigma_1$ for reference under Non-Relational Difference in Table 4.

The relational error results in Table 4 mirrors the extrapolation performance confirming that representation quality is important for effective generalization. High relational error compounds over longer word lengths hindering generalization whereas low relational error allows MatrixNet to automatically generalize to longer word lengths through invariance to group relations. These results show that MatrixNet, particularly the MatrixNet-LN and MatrixNet-NL variants, is able to learn group representations invariant to the group relations allowing for effective generalization to longer unseen group words.

Table 4: Relational error of MatrixNet models trained on length extrapolation dataset. The non-relational difference is computed between two non-equivalent braids for comparison. High relational error compounds for longer words resulting in poor extrapolation.

| MatrixNet Variation | Relational Error | Non-relational Difference |
|---|---|---|
| MatrixNet | $13.63 \pm 6.83$ | $33.64 \pm 11.28$ |
| MatrixNet-MC | $2.58 \pm 2.10$ | $9.60 \pm 2.5$ |
| MatrixNet-LN | $0.071 \pm 0.018$ | $4.96 \pm 0.55$ |
| MatrixNet-NL | $0.066 \pm 0.009$ | $4.99 \pm 0.45$ |

## 5.3 Visualizing the Learned Representations

We present some visualizations of the learned representations of the braid group from the highest performing variant, MatrixNet-NL. Figure 2 shows visual plots of the learned representations. In

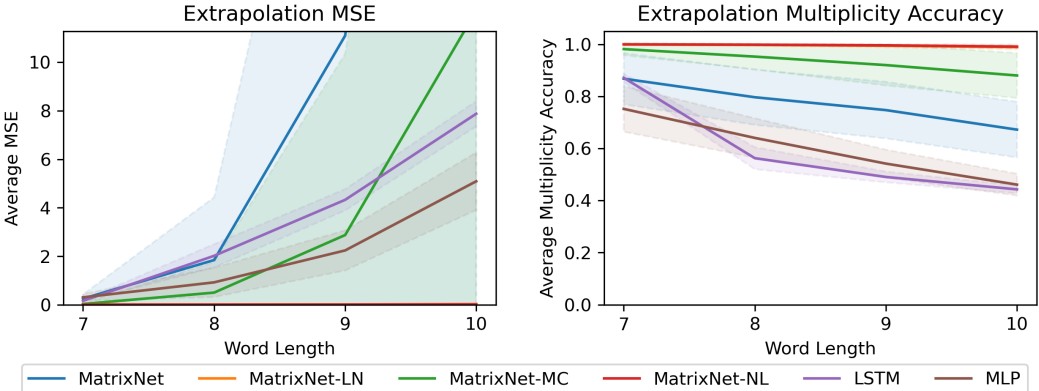

Figure 2: Length extrapolation results. Left: The plot shows how MSE grows for increasing word lengths ($y$-axis is truncated for clarity). Right: The plot shows how the average accuracy decays for increasing word lengths. The relatively high accuracy of MatrixNet and MatrixNet-MC compared to baselines suggests that the high MSE is caused by outliers with multiplicity predictions much higher than the ground truth.

the last two plots, the learned representation for two equivalent words are approximately equal even though this relation is not among the relations $r_j$ used in the loss $\mathcal{L}_{rel}$. This shows MatrixNet does indeed generalize over relations, allowing it to generate nearly identical representations for equivalent words.

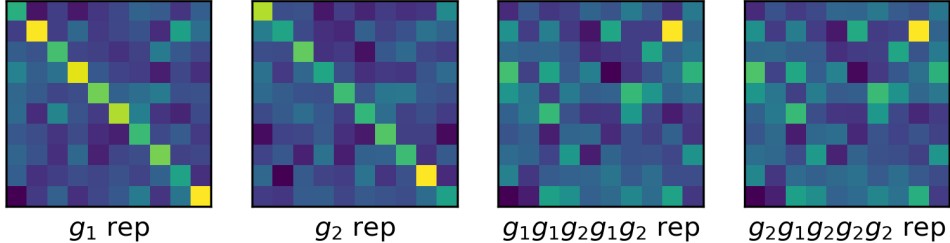

Figure 3: Visualization of learned matrix representations. The first two figures show the representations for the generators of $B_3$. The last two figures show the representation for equivalent words that are generated by the relations of $B_3$.

## 6   Conclusion

In this paper we have presented MatrixNet, a novel neural network architecture designed for learning tasks with group element inputs. We developed 3 variations of our approach which structurally enforce group axioms and a regularization approach for enforcing invariance to relations. We evaluate MatrixNet on two group learning problems over several finite groups and $B_3$ and demonstrate our model's performance and ability to automatically generalize to unseen group elements. In future work we plan to develop interpretability analysis methods based on group representation theory to better understand the structure of MatrixNet's learned representations. Understanding the learned representations may provide valuable insights and explanations of the model outputs assisting with generating new conjectures for open mathematical research problems.

**Limitations**   The current work relies on the assumption that the studied group is finitely presented which limits us to discrete groups. However, learned group representations may also be useful for learning over Lie groups. In such case, extending our method will require working with infinitesimal Lie algebra generators. Additionally, while the group axioms are strictly enforced by the model structure, the fact the relations are enforced using auxiliary loss terms means the homomorphism property is not exact. Future work may explore methods of reducing this error.

**Acknowledgements** This work is supported in part by NSF 2134178. Lucas Laird is supported by the MIT Lincoln Laboratory Lincoln Scholars program. Circe Hsu is supported by a Northeastern University Undergraduate Research and Fellowships PEAK Experiences Award. The authors would like to thank Mustafa Hajij and Paul Hand for helpful discussions.

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

# A   More details about the categorical braid group action

The aim of this appendix is to provide a few more details about the particular categorical braid group action that we use in our experiments.

## A.1   Sketch of the construction of the category

The category $\mathcal{C}_n$ we consider is the 2-Calabi–Yau triangulated category associated to the Dynkin graph of type $A_n$. This is an undirected graph with $n$ vertices and $n-1$ edges arranged in a line, as shown in Figure 4.

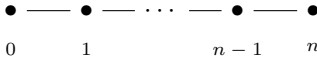

Figure 4: The Dynkin graph of type $A_n$.

Let $\Gamma_n$ be the Dynkin graph of type $A_n$. Let $\Gamma_n^{\mathrm{dbl}}$ be its *doubled quiver*, which is a directed graph in which each undirected edge of $\Gamma$ is replaced by a pair of oppositely oriented directed edges, as shown in Figure 5.

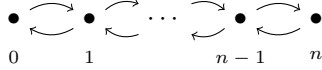

Figure 5: The doubled quiver $\Gamma_n^{\mathrm{dbl}}$ of the Dynkin graph of type $A_n$.

Recall that the *path algebra* of a directed graph (or quiver) $Q$ over some field $k$ is generated as a free vector space by all possible paths in $Q$, including the trivial paths at each vertex. The product in $kQ$ is given as follows: let $q \colon a \to b$ be a path and $p \colon c \to d$ be a path. The product $pq$ is equal to zero unless $b = c$. If $b = c$, then the product $pq$ is simply the composite path $a \xrightarrow{q} b \xrightarrow{p} d$. The path algebra of $Q$ is denoted $kQ$.

We are interested in a quotient of the path algebra of $\Gamma_n^{\mathrm{dbl}}$ called the zig-zag algebra, and denoted $Z_n$. To obtain $Z_n$, we impose the following relations on $k\Gamma_n^{\mathrm{dbl}}$. In what follows, $(i|i \pm 1)$ represents the unique arrow $i \to i \pm 1$.

- For each $i$, set
$$(i+1|i)(i+2|i+1) = 0 \text{ and } (i-1|i)(i-2|i-1) = 0.$$

- For each $i$, set
$$(i+1|i)(i|i+1) = (i-1|i)(i|i-1).$$

Consequently, in the zig-zag algebra, any path of length at least 3 is automatically zero, and the only surviving paths of length 2 are back-and-forth loops starting from any vertex (and all possible such loops are set to be equal). The paths of length 0 and 1 remain as-is.

Let $Z_n - \mathrm{proj}$ be the category of (graded) projective modules over $Z_n$. The category $\mathcal{C}_n$ is constructed as a differential graded version of the bounded homotopy category of complexes of projective modules over $Z_n$, in which we identify the internal grading shift with the homological grading shift, and consequently also the triangulated shift. For more details, see, e.g. [44, Section 6].

## A.2 Important properties of the category

In this section, we record some important properties of the category $\mathcal{C}_n$. First, the category $\mathcal{C}_n$ is *triangulated*: there is a a triangulated shift functor $[1] \colon \mathcal{C}_n \to \mathcal{C}_n$ which is an equivalence. An $n$-fold composition of $[1]$ is denoted $[n]$.

Denote by $\mathrm{Hom}(A, B)$ the set of morphisms in $\mathcal{C}_n$ from an object $A$ to an object $B$. Sometimes we write $\mathrm{Hom}(A, B)$ as $\mathrm{Hom}^0(A, B)$, and further write $\mathrm{Hom}^n(A, B)$ to mean $\mathrm{Hom}(A, B[n])$ for any integer $n$.

The category $\mathcal{C}_n$ is generated as a triangulated category by the objects $P_1, \dots, P_n$. That is, the smallest triangulated subcategory of $\mathcal{C}_n$ (closed under isomorphisms) that contains the objects $P_1, \dots, P_n$ is $\mathcal{C}_n$ itself. These objects correspond to the indecomposable projective modules of the zig-zag algebra $Z_n$.

Each object $P_i$ is *spherical*. This means that
$$\mathrm{Hom}^n(P_i, P_i) = \begin{cases} k & n = 0 \text{ or } n = 2, \\ 0 & \text{otherwise.} \end{cases}$$

**Remark A.1.** The reason for the notation is that the ring of endomorphisms of $P_i$ of all possible degrees is isomorphic to the cohomology ring of a sphere (in this case a 2-sphere).

It is a general fact that any spherical object $X$ of a triangulated category gives an associated functor $\sigma_X$, called the *spherical twist* in $X$ (see Seidel–Thomas [45] for more details.) The functor $\sigma_X$ is an auto-equivalence; that is, it has an inverse equivalence $\sigma_X^{-1}$ such that compositions in both directions are isomorphic to the identity functor.

In particular, we obtain equivalences $\sigma_{P_i} \colon \mathcal{C}_n \to \mathcal{C}_n$.

## A.3 The Jordan–Hölder filtration

The category $\mathcal{C}_n$ has a bounded $t$-structure. This means that there is an abelian subcategory $\mathcal{A}_n \subset \mathcal{C}_n$, such that every object $X \in \mathcal{C}_n$ has a unique finite filtration whose factors lie in $\mathcal{A}[i]$ for decreasing values of $i$. This filtration is called the *cohomology filtration*. In fact, this abelian subcategory $\mathcal{A}_n$ is also generated by the objects $P_i$: it is the extension-closure of the objects $P_i$. Moreover, the objects $P_i$ are simple objects of $\mathcal{A}_n$.

First consider any object $X \in \mathcal{A}_n$. The category $\mathcal{A}_n$ is a finite-length abelian category. It is a standard fact that $X$ has a Jordan–Hölder filtration whose factors are simple objects in $\mathcal{A}_n$, namely the objects $P_i$.

Now consider a general object $X \in \mathcal{C}_n$. We first consider the cohomology filtration of $X$, with factors $Y_j \in \mathcal{A}[j]$. For each $Y_j \in \mathcal{A}[j]$, we consider its (shifted) Jordan–Hölder filtration, which breaks $Y_j$ up further into copies of $P_i[j]$. Putting these two together, we obtain a finer filtration of the object $X$, which we also call the Jordan–Hölder filtration of $X$.

The factors of this Jordan–Hölder filtration are shifted copies of $P_i$ for all $i$. Thus we can count the number of occurrences of each $P_i$ in the Jordan–Hölder filtration of $X$, and it is well-known that these counts do not depend on the specific choice of the Jordan–Hölder filtration.

## A.4   The action of the braid group

Recall the spherical twist functors $\sigma_{P_i} : \mathcal{C}_n \to \mathcal{C}_n$. A remarkable observation of Seidel–Thomas [45] is that these functors (weakly) obey the relations of the $n$-strand braid group. That is, we have isomorphisms of functors
$$\sigma_{P_i}\sigma_{P_j} \cong \sigma_{P_j}\sigma_{P_i}$$
whenever $|i - j| \neq 1$, and
$$\sigma_{P_i}\sigma_{P_j}\sigma_{P_i} \cong \sigma_{P_j}\sigma_{P_i}\sigma_{P_j}$$
whenever $|i - j| = 1$.

Since these are precisely the relations of the group $B_n$, we obtain an action of $B_n$ on the objects of the category $\mathcal{C}_n$.

## A.5   Open problems and future directions

Consider the following broad question. Given an object $X$ of $\mathcal{C}_n$ and a braid $g \in B_n$, can we relate the Jordan–Hölder multiplicities of $\beta(X)$ to the Jordan–Hölder multiplicities of $X$? Stated more simply, can we compute the Jordan–Hölder multiplicities of $\beta(P_i)$ for any $i$ and any $\beta$?

Answers to this question are known in some cases, and remain open in others. For instance, a complete answer was obtained by Rouquier–Zimmermann [46] for the 3-strand braid group $B_3$ acting on $\mathcal{C}_3$. This answer was rediscovered and refined in terms of more general filtrations (Harder–Narasimhan filtrations) in [44] and [47].

It is also known that if $\beta = \sigma_{P_j}^\ell$ for some $\ell$, then the limit as $\ell \to \infty$ of the counts of $\beta(X)$ can be obtained, up to a common scaling factor, by computing the sum of the dimensions of $\mathrm{Hom}^m(P_j, X)$ for all $m$ [47].

However, for the vast majority of values of $n$ and most of the elements of the braid groups $B_n$, we do not have a good answer to this question. While there is vast potential for future work, we write down a few specific open problems.

1. Generalise the Rouquier–Zimmermann theorem (and its corresponding versions in [44] and [47]) to larger values of $n$.

2. We can compute a finer version of Jordan–Hölder multiplicities: split up the number of occurrences of each $P_i$ by degree shift. That is, record the number of occurrences of $P_i[d]$ separately for every possible $d$. This information can be encoded in a polynomial in one variable in $q^{\pm 1}$, in which the coefficient of $q^d$ is the multiplicity of $P_i[d]$.

   Generalise the Rouquier–Zimmermann theorem in this setting to larger values of $n$.

3. By using a more refined version of Jordan–Hölder multiplicities, known as Harder–Narasimhan multiplicities, we observe that the possible Harder–Narasimhan factors of any object of the form $\beta(P_i)$ are highly constrained, and satisfy some very nice combinatorial properties.

   This constraint can be explicitly described for any $B_n$ via a geometric model (due to Khovanov–Seidel [48]) for objects in the category $\mathcal{C}_n$. Nevertheless, the relationship of these constrained sets with the action of $B_n$ is mysterious. For example, given an object $X$, is there an algorithm to write down a braid that will send $X$ to an object with a desired set of Harder–Narasimhan filtration factors?

4. Can we use the combinatorial structure mentioned above to algorithmically write down combinatorial actions of braid groups on simpler sets? What properties do these actions satisfy?

5. All of the categorical constructions described in this paper also go through for more general versions of braid groups, known as Artin–Tits braid groups. All of the questions above remain open for all but the simplest cases of Artin–Tits groups.

# B  Dataset and Model Parameter Details

## B.1  Symmetric Group Dataset

We generated a dataset of $500,000$ samples consisting of words of the free group $F_{10}$, and labels corresponding to their order as elements of $S_{10}$. The first step of dataset generation was to fix a maximum word length chosen such that it is possible to sample every element of $S_{10}$. For a generating set corresponding to adjacent transpositions of elements in $S_n$, this longest word will be of length $\frac{n(n-1)}{2}$ [49], and for $S_{10}$ choose our maximum length to be 64. We define a uniform distribution on the set of generators $\sigma_0, \sigma_1, ..., \sigma_{n-1}$ where $\sigma_0 = 1$ and all other $\sigma_i = (i\ i+1)$. Informally, our generating set consists of adjacent transpositions of the form $(i\ i+1)$ along with an identity generator. The presence of the identity generator adds variability to word length while enforcing identity invariance. Sample order labels in $S_{10}$ are computed using the SymPy package [50].

While we do not strictly enforce a separation of elements between training, test, and validation sets, it is statistically unlikely to have any significant overlap between the splits. Recall that $|S_{10}| = 10! = 3,628,800$. Our dataset of $500,000$ samples therefore covers at most $13.7\%$ of $S_{10}$, implying the likelihood of significant overlap between partitions upon reduction is very low. Moreover, because we are sampling *unreduced* words from $F_{10}$ of length 64, there are $10^{64}$ possible words we could sample from, making the probability of direct overlap between partitions effectively zero.

## B.2  Categorical Braid Action Experiment Details

**Dataset Generation**  An initial dataset of Jordan–Hölder multiplicities for braid words up to length 6 was provided. We implemented a state automaton algorithm from [47] to generate additional examples for longer braid words. This method was compared against the Jordan–Hölder multiplicities of the initial dataset to verify correctness.

**Baseline Comparison Experiment Details**  The baseline comparison dataset consists of 47,831 examples with braid words up to length 8. The data was split into $60\%$ training data, $20\%$ validation data, and $20\%$ test which were fixed for all models. All of the models trained using an Adam optimizer with a learning rate of 1e–4 and a batch size of 128. The chosen parameters for the models are:

- MatrixNet: Single channel $14 \times 14$ matrix size
- MatrixNet-LN: Single channel $10 \times 10$ matrix size, 128 dimensions for linear network in the matrix block
- MatrixNet-MC: 3-channel $8 \times 8$ matrix size
- MatrixNet-NL: Single channel $10 \times 10$ matrix size, 128 hidden dimensions and a tanh non-linearity between linear layers of matrix block
- MLP: 3-layer MLP with 128 hidden dimensions for each layer and ReLU activation functions followed by a single linear layer output.
- LSTM: 6 LSTM layers with 16 dimensional input embeddings and 32 hidden dimensions followed by a 2-layer MLP classifier with 64 hidden dimensions and ReLU activation.
- Transformer: 3 transformer layers with 4 attention heads, 16 dimensional embeddings and 32 hidden dimensions. Used mean pooling and a single linear layer output.

All of the MatrixNet architectures used a 2-layer MLP with 128 hidden dimensions and ReLU activation to compute output after the matrix block.

**Length Extrapolation Experiment**    For the generalization experiment we generated a dataset of all braid words up to length 7 which was split into $80\%$ training and $20\%$ validation. We also generated three separate test datasets of 10,000 examples each with braid words of length 8, 9, and 10 to evaluate how performance degrades over increasing length. The models were trained for 100 epochs on the training and validation sets and then tested on the three test sets.

**Regularization Details**    All MatrixNet architectures were trained using the regularization loss defined in Section 4.3. We chose the Frobenius norm for the norm and used the braid relation $\sigma_1\sigma_2\sigma_2 = \sigma_2\sigma_1\sigma2$ and the inverse $\sigma_1^{-1}\sigma_2^{-1}\sigma_1^{-1} = \sigma_2^{-1}\sigma_1^{-1}\sigma_2^{-1}$. The regularization term was added to the loss every 10 training batches.

**Hardware Details**    All of the categorical braid action experiments were run on a machine with a single Nvidia RTX 2080 ti GPU.

## B.3   Length Interpolation Results

We also performed a length interpolation generalization experiment to test how well each model generalizes to braid words that are shorter than the maximum length seen during training. The results are not presented in the main body of the paper as all models generalize to shorter braid words.

Table 5: Length 5 interpolation performance of baseline models and MatrixNet variations. Test MSE and accuracy is measured over test set which contains braid words of same length as training.

| Model Type | Test MSE | Test count acc. | Interpolation MSE | Interpolation count acc. |
|---|---|---|---|---|
| MLP | 0.3628 | 72.2% | 0.174 | 80.4% |
| LSTM | 0.7995 | 53.2% | 0.5442 | 57.7% |
| MatrixNet | 1.097 | 59.7% | 0.413 | 67.3% |
| MatrixNet-LN | 0.0120 | 99.9% | 0.0077 | **100**% |
| MatrixNet-MC | 0.2544 | 91.1% | 0.0691 | **100**% |
| MatrixNet-NL | **6.8e–3** | **100**% | **3.8e–3** | **100**% |

