# OpenReview forum: "MatrixNet: Learning over symmetry groups using learned group representations"
_NeurIPS.cc/2024/Conference — NeurIPS 2024 poster_

### Official Review · Reviewer_puxG · 2024-06-30

**Soundness:** 2
**Presentation:** 2
**Contribution:** 2
**Rating:** 5
**Confidence:** 4

**Summary:**

In this paper, authors study the question of what feature representations to use for learning tasks with inputs coming from a symmetry group. They propose MatrixNet, a neural network architecture that learns matrix representations of group element inputs instead of using predefined representations.

The main contributions are as follows:
1. Formulation of the problem of learning over groups as a sequence learning problem in terms of group generators with invariance to group axioms and relations.
2. The proposed neural network architecture, MatrixNet, achieves higher sample efficiency and generalization over several more general baselines in prediction tasks over the symmetric group and Artin braid group.
3. The matrix block method for constraining the network to respect the group axioms and an additional loss term for learning group relations.

**Strengths:**

The strength of the paper is building up links among group reprsentation theory, learning tasks and neural networks, more specifically:
1. Formulation of the problem of learning over groups as a sequence learning problem in terms of group generators with invariance to group axioms and relations.
2. The matrix block method for constraining the network to respect the group axioms and an additional loss term for learning group relations.

**Weaknesses:**

The motivation is not so clear to me. The applications of group theory are everywhere in the real-world. Thus, how to choose a good representation for the group which is associated with the learning task is more important. In this paper, authors formulate this problem but do not give a clear answer (see my questions below), and authors focus on solving a more abstract group theory question instead which is already studied a lot by mathematician.

I didn't see any insight behind the paper. For the experiment, I think it only solved a mathematical problem at simple setting which is far from what we expect. For example, how the model works when the group's order become larger?

**Questions:**

1. The motivation is to solve a mathematical problem or real-world learning task? I hope it can help me understand the paper clearly and inprove the writing.

2. In subsection 4.1, authors formulate the problem. In line 166, can $f$ be seen as a representation of $G$ or not? If it is a representation, can you write a explicite expression as an example? Since if it is a representation, the target domain $R^{c}$ may not be a linear space which makes it unclear for me.

3. Can you show more experiment results? Such as larger symmetric group? Also I didn't understand the meaning of predict the order of the group element? I think we should determine it instead of estimating it? Maybe I did't get your point, but I am open to discuss.

**Limitations:**

see the weaknesses and questions.

---

> ### Author Rebuttal · Authors · 2024-08-07
>
> ## Response to Reviewer puxG
> > The motivation is not so clear to me. The applications of group theory are everywhere in the real-world. Thus, how to choose a good representation for the group which is associated with the learning task is more important. In this paper, authors formulate this problem but do not give a clear answer.
>
> In the $S_{10}$ experiment in section 5.1 we compare against using precomputed representations and show significantly worse performance compared to the learnable representations of MatrixNet. This ablation shows MatrixNet can automatically learn more useful group representations for a given task. Our results parallel other results in deep learning showing that learned feature representations often outperform expert-engineered features. We discuss our motivation below with your question.
>
> > I didn't see any insight behind the paper. For the experiment, I think it only solved a mathematical problem at simple setting.
>
> The task used for our experiment on the Artin braid group in section 5.2 is an open mathematical research problem. This task is limited to $B_3$ since the multiplicity counts are not known for any larger braid groups. The experiment over $S_{10}$ in section 5.1 was chosen to provide a well-studied group with known representations to compare precomputed representations against the learned representations of MatrixNet.
>
> __Questions__:
>
> __Q1__: The motivation is not so clear to me. The motivation is to solve a mathematical problem or real-world learning task?
>
> __A1__: Thank you for the feedback. We will make sure to make the motivation and application more clear in the paper. The second task used in our experiments over the Artin braid group is a current open mathematics research problem. Our goal is to create a model that can help mathematicians build intuition and formulate conjectures by 1) computing additional data points and 2) providing new insights through inspecting the learned representations. We believe this application is relevant to real mathematical research.
>
>
>
> __Q2__: In line 166, can $\mathcal{f}$ be seen as a representation of \mathcal{G} or not?
>
> In line 166 the function $\mathcal{f}$ does not constitute a representation of the group since the output space does not obey the group composition operation. Let $v_i = \mathcal{f}(g_i)$ be the vector target for a group element. Then if $g_k = g_i \circ g_j$ it is not the case that $v_k = v_i + v_j$.
>
>
> __Q3.1__: Can you show more experiment results? Such as larger symmetric group?
>
> __A3.1__: Yes, thank you for the suggestion. Using the same data generation scheme and data splits detailed for the $S_{10}$ experiment, we generated 800,000 samples for the group $S_{12}$. $S_{12}$ is roughly two orders of magnitude larger than $S_{10}$ (12! vs 10! elements). MatrixNet achieves an MSE of 3e-4 across all dataset splits with a classification accuracy of over 99%. The only increase to the network size necessary in this case was altering the rep size from 10 to 12.
>
> __Q3.2__: Also I didn't understand the meaning of predict the order of the group element? I think we should determine it instead of estimating it?
>
> __A3.2__: Sorry for confusing terminology.  We will change “predict” to “determine.”  Our model outputs an exact order, not an estimate.  We used the term predict in a colloquial way to refer to the model output.

---

> > ### Comment · Reviewer_puxG · 2024-08-09
> >
> > Thanks for authors' reply. It helps me understand the paper better.

---

### Official Review · Reviewer_KZHf · 2024-07-03

**Soundness:** 3
**Presentation:** 3
**Contribution:** 3
**Rating:** 7
**Confidence:** 4

**Summary:**

The paper describes MatrixNet, a method to learn group representations such that they are optimized for a certain task of interest using a neural network. , The neural network takes in a group element in the form of a sequence of generators that compose to form the group element and forms an intermediate output that contain the matrices that are the group representations for each of the generators. The generators can them be multiplied to form the group representation which is then fed to further neural layers that are task-specific. The network is trained so as to (approximately) satisfy the constraints of group representations using architectural constraints and the loss function. Experiments on predicting the order of a element of the symmetric group and Jordan-Holder multiplicities for the 3-strand braid group.

**Strengths:**

1. The paper present new ideas for learning on groups. Design of architectures is clever and done well with good mathematical justification. The architectures provably output group representations.

2. Experiments show that the proposed architecture leads to learning group representations that lead to better results compared to just using

3. The paper also does a good study of the possible variations of MatrixNet and presents experimental results for them.

**Weaknesses:**

1. One of the weaknesses is that the parts about the braid group are very difficult to follow and probably need a lot more background for readers and attendees for this conference. This includes myself and I was not able to fully follow the details or why that experiment is important.

2. The experiments are also a little weak in my opinion. It was not clear to me how the learned representations were different from precomputed ones and why they were better for a given task. Also, what happens when the size of the group representations is larger. Perhaps for the first experiment, it would be nice to see the results as a function of the size of the group representation. I am not sure if the proposed architecture and learning mechanism can learn good representations when the size is large. This is important to address, in my opinion.

**Questions:**

No additional questions.

**Limitations:**

Yes, authors have listed the limitations of the current submission and suggested directions for future research.

---

> ### Author Rebuttal · Authors · 2024-08-07
>
> ## Response to Reviewer KZHf
> > Parts about the braid group are very difficult to follow and probably need a lot more background for readers and attendees for this conference.
>
> Thank you for this feedback. We will make sure to revise this section to add more clarity. Intuitively the braid group defined in the paper represents all of the possible ways to braid a set of $N$ ropes. The braid group is closely connected to many fields of math but in particular we are interested in how it acts on mathematical categories through “categorical braid actions”. Mathematicians are interested in studying “filtrations” of the objects of categories but there isn’t a unique filtration for all objects within a category. The categorical braid actions however act on all such filtrations in the same way and so the multiplicity counts provide a canonical way to describe object filtrations regardless of the specific filtration. We include a more in-depth discussion of the braid group and categorical braid actions in section A of the appendix.
>
> > The experiments are a little weak in my opinion. It was not clear to me how the learned representations were different from precomputed ones and why they were better for a given task.
>
> Our results parallel other results in deep learning showing that learned feature representations often outperform expert-engineered features. The precomputed representations we used are a natural choice for matrix representations for the groups used. For example, the group $S_{10}$ represents all of the ways to permute 10 objects and can intuitively be represented by 10x10 permutation matrices, which is the precomputed representation we use in the experiment. However, depending on the task, this may not be the most useful representation. For example, the sign representation of a permutation is simply $\rho(\sigma) = \pm1$ depending on the parity of $\sigma$. This would be a useful feature for determining a group element's order since every permutation of odd order has even parity. Our approach is designed to automatically learn a representation that is useful for the given task.
>
> > What happens when the size of the group representations is larger. I am not sure if the proposed architecture and learning mechanism can learn good representations when the size is large.
>
> Our method can scale to large representations using MatrixNet-MC. MatrixNet-MC assumes large representations have a block diagonal structure, which is efficient since it means the number of trainable parameters grows asymptotically linearly instead of quadratically. For well chosen block sizes, this does not harm expressivity since for many groups, all representations will be block diagonal with respect to a good choice of basis with the maximum block size equal to the largest dimensional irreducible representation. Even for very large groups these irreducible representations are low dimensional which can be learned by MatrixNet-MC.
>
> We omitted results with larger representation sizes as the performance did not change with matrix size. We include additional results for MatrixNet on the braid group task with doubled representation sizes in Table 2.
>
> | Model | MSE | Acc. |
> | ----------- | ---- | ---- |
> | MatrixNet | 0.975 | 85% |
> | MatrixNet-MC | 0.052 | 96% |
> | MatrixNet-LN | 4.5e-4 | 100% |
> | MatrixNet-tanh | 1.1e-3 | 100% |

---

> > ### Comment · Reviewer_KZHf · 2024-08-13
> > **Thank you for the response**
> >
> > Thank you for answering my questions.
> >
> > I believe the authors have done a good job in addressing most of the concerns that all the reviewers had.
> >
> > The authors should definitely include all the new results in the rebuttal into the main paper.
> >
> > I will raise my score to a 7.

---

### Official Review · Reviewer_JC7A · 2024-07-09

**Soundness:** 3
**Presentation:** 3
**Contribution:** 2
**Rating:** 4
**Confidence:** 3

**Summary:**

This paper studies feature representations of a group element for supervised learning. It considers a regression task where an input is a group element g of a finite group and a target is some label. Firstly, g is decomposed into a sequence of generators (g_1, ..., g_n) such that g = g_1 \circ ... \circ g_n. Next, each generator is mapped by a trainable matrix embedding W: Gen(G) --> R^{n \times n} followed by the matrix exponential so that we can get a group representation M_i = exp(W(g_i)) ∈ GL(n). Then their product M_1 \circ ... \circ M_n becomes the feature. Similar variants are also proposed. The performance is evaluated on two synthetic tasks: the order prediction of the symmetric group S_10 and the action prediction of the Braid group B_3.

**Strengths:**

1. The paper is well written. Technical details are clear enough.
1. The idea to construct a feature representation of a group element as a learnable parameter is interesting and it is a possibly promising direction.

**Weaknesses:**

1. Limited applicability to real tasks. To the best of my knowledge, regression (or classification) of group elements is practically important for continuous groups such as SO(3) (e.g., pose estimation). However, the current approach is only applicable to finite groups. Also, I think the tasks conducted in the experiments are not directly bridged to real problems, and I feel uncertain about how the proposed method is practically valuable.
1. It is unclear which component significantly contributed to the final performance gain. The proposed method consists of (at least) two parts: the decomposition of g into generators and a trainable representation of a generator. The current experiments are not designed to evaluate them separately.
1. The experiments are not convincing enough—they are relatively small scale, use synthetic tasks only, and have less variety (two tasks).

**Questions:**

1. Given g, is its decomposition into the generators always uniquely determined?
1. In the experiment of 5.1, what will happen when we employ the group decomposition while using the fixed representation? I mean, the feature of g is given as concat[\rho(g_1), ..., \rho(g_n)] where g = g_1 \circ ... \circ g_n and \rho(g_i) is some representation of g_i (e.g. irreducible rep).

Typo:
* One of M_{g_ik^-1} would be {M_{g_ik}}^{-1} in the equation below line 203.

**Limitations:**

Limitations are adressed.

---

> ### Author Rebuttal · Authors · 2024-08-07
>
> ## Response to Reviewer JC7A
> > Limited applicability to real tasks. The current approach is only applicable to finite groups.
>
> Our method is **not** limited to finite groups. One of our experiments focuses on the infinite Artin braid group. You are correct that our approach is not formulated for continuous groups as it is limited to **discrete** groups.
>
> > Also, I think the tasks conducted in the experiments are not directly bridged to real problems, and I feel uncertain about how the proposed method is practically valuable.
>
> The second task used in our experiments over the Artin-Braid group is a real current research problem in pure math. Mathematicians have only found a way to compute the answer for the simplest braid group $B_3$ – and they do not have a simple or intuitive formula. Our goal is to create a model that can help mathematicians build intuition and formulate conjectures by 1) computing additional data points and 2) providing new insights through inspecting the learned representations. We believe this application is relevant to real mathematical research.
>
> > It is unclear which component significantly contributed to the final performance gain: decomposition of g into generators or a trainable representation of a generator.
>
> We do test the impact of a tranable representation independent of the decomposition in our first experiment in section 5.1 of the paper by comparing against the precomputed representation. This ablation replaces the learnable representations with the permutation representation of $S_{10}$ and we see significantly worse performance when compared to learnable representation.
>
> We also show that just decomposing g into generators but not using a learned or fixed group representation does not result in improved performance. We compare against two sequential baselines, a transformer and LSTM, which both take the decomposition of g as input. These models however do not learn a group representation showing that just the decomposition of g into a generator sequence does not explain the improved performance of MatrixNet.
>
> > The experiments are not convincing enough—they are relatively small scale, use synthetic tasks only, and have less variety (two tasks).
>
> We disagree that only synthetic tasks are used. The task used in our Artin braid group experiment is an unsolved math problem from a recent pure mathematics publication [39]. Mathematicians have only found a way to compute the answer for the simplest braid group B_3 – and they do not have a simple or intuitive formula. Even so, we believe the two groups used, $S_{10}$ and $B_3$, are sufficiently large with $|S_{10}| = 10!$ and $B_3$ being an infinite group.
>
> __Questions__:
> __Q1__: Given g, is its decomposition into the generators always uniquely determined?
>
> __A1__: No, the decomposition of g is not unique. This is a great question and is a huge part of the motivation behind our approach. MatrixNet is designed to be invariant to the choice of decomposition.
>
> __Q2__: In the experiment of 5.1, what will happen when we employ the group decomposition while using the fixed representation? I mean, the feature of g is given as concat[$\rho(g_1)$, ..., $\rho(g_n)$] where $g = g_1 \circ ... \circ g_n and \rho(g_i)$ is some representation of $g_i$ (e.g. irreducible rep).
>
> __A2__: You could do this, but concatenating in this way means the feature size grows with the length of the decomposition. Since the decomposition of is not unique, another drawback is that this method would result in different features for two different but equivalent decompositions of g. The length of a decomposition is also unbounded. For example, in the braid group an element can be decomposed into arbitrarily many generators which would result in unbounded feature sizes. Our method by contrast is more efficient since feature sizes are fixed and with low relational error produces unique representations for a group element.
>
> __Typo__: One of $M_{g_ik^-1}$ would be ${M_{g_ik}}^{-1}$ in the equation below line 203.
>
> Good catch. We will fix it.

---

> > ### Comment · Reviewer_JC7A · 2024-08-14
> > **To authors**
> >
> > Thank you for your response. My concerns are almost addressed. Still, I keep the original score because I'm not convinced enough of its applicability. Since I'm not a mathematician and I cannot judge how the group problems are significant.

---

### Official Review · Reviewer_jQuD · 2024-07-11

**Soundness:** 3
**Presentation:** 3
**Contribution:** 3
**Rating:** 6
**Confidence:** 3

**Summary:**

The authors use neural networks to learn group representations. A group element is represented by its generators formatted as a sequence of learned matrix representations. These generators are mapped to a single matrix representation of the group element via the Matrix Block which enforces group axioms. The resulting feature is used for downstream tasks on groups. The authors show that the proposed architecture MatrixNet successfully predicts group element orders for S_10 and Jordan-Holder multiplicities for braid group B_3. Further analysis on word length extrapolation and visualization demonstrates the superiority and usefulness of the approach.

**Strengths:**

1. In Table 1, the faster convergence of MatrixNet potentially indicates that the proper architectural constraints provide good inductive bias for learning group representations and solving downstream tasks over groups. It’s good to see how one can explicitly build in these constraints in the problem of group representations and that it works better than a less constraining MLP without domain-specific inductive bias.
2. The paper is well-written.

**Weaknesses:**

1. It’s unclear why naive MatrixNet and MatrixNet-MC cannot extrapolate to longer word length.
2. Empirical evaluations are a bit limited. It might be helpful to evaluate groups with different properties (see questions).

**Questions:**

In the matrix block, is it possible to introduce commutative operations between different matrix representations of group generators if the input group element comes from an abelian group. It seems possible to introduce further architectural constraints for specific groups.

---

> ### Author Rebuttal · Authors · 2024-08-07
>
> ## Response to Reviewer jQuD
> >It’s unclear why naive MatrixNet and MatrixNet-MC cannot extrapolate to longer word length.
>
> While MatrixNet and MatrixNet-MC underperform compared to our other two variants, it is overstated to say they cannot extrapolate to longer word lengths. Despite their high MSE, both approaches maintain relatively high accuracy compared to our baselines. That said, MatrixNet-LN and MatrixNet-tanh have better extrapolation. The reason for this performance discrepancy is that MatrixNet and MatrixNet-MC have higher relational error indicating they do not learn group representations as accurately. This error compounds for longer words. To help make this difference clearer we computed the relational errors of the different models on the Artin Braid group, shown in Table 1, which we will add to the paper.
>
> | Model | Rel. Error |
> | ------------ | -------- |
> | MatrixNet | 14.78 |
> | MatrixNet-MC | 5.21 |
> | MatrixNet-LN | 0.33 |
> | MatrixNet-tanh | 0.45 |
>
> > Empirical evaluations are a bit limited. It might be helpful to evaluate groups with different properties.
>
> Thank you for the feedback. The task we used for the braid group was a primary motivation for our approach. We are limited to $B_3$ since multiplicity counts for larger braid groups are not known. We chose the symmetric group for our initial experiment in section 5.1 as it is a well-studied group that has the unique property that all finite groups are isomorphic to a subgroup of a symmetric group. It also is closely connected to the braid group making it well suited for ablation tests. We believe that these two groups provide a strong foundation for evaluating MatrixNet but we will perform further evaluations on an abelian group as well.
>
> __Questions__:
>
> __Q1__: In the matrix block, is it possible to introduce commutative operations between different matrix representations of group generators if the input group element comes from an abelian group?
>
> __A1__: Yes this is possible. For an abelian group commutativity can be enforced in two ways in MatrixNet. One is with a loss term $L = |M_1 M_2 - M_2 M_1|$ and the other is by choosing the learned matrix representation to be diagonal, i.e. a direct sum of one dimensional representations. Since irreps for an abelian group are one dimensional, this would not harm expressivity of the learned representation and would enforce exact commutativity. We did not use diagonal matrices for the non-abelian groups specifically for this reason. More concretely, MatrixNet-MC with scalar channels is an architecture that is constrained to learn commutative representations.

---

> > ### Comment · Reviewer_jQuD · 2024-08-09
> >
> > Thank you for the additional experiments and clarifications on how to incorporate other group constraints. My concerns are addressed. I maintain my score.
> >
> > Reasons for not raising my scores: I am not familiar with abstract algebra enough to see the full implication of this work. As of now the proposed approach works well on tasks chosen in the paper. It's hard for me to see if the specific architectural constraints here generalize to broader classes of groups.

---

> > > ### Author Response · Authors · 2024-08-12
> > >
> > > Thank you for your comment - To test Matrixnet performance on broader classes of groups we've generated data for the following groups: $S_{12}$, $S_{5} \times S_{5} \times S_{5}\times S_{5}$, and $C_{11} \times C_{12} \times C_{13} \times C_{14} \times C_{15}$, and trained using Matrixnet-tanh with minimal tuning. We hope that the inclusion of the $S_5$ product group and latter Abelian group help illustrate the robustness of our method. Results are summarized in the table below:
> > >
> > >
> > > | Group| Rep Size | Loss | Test Acc |
> > > |  :----------------  | :-: | :------:  |  :----:  |
> > > | $S_{12}$ | 12 | 1.1e-2 | 98.4%
> > > | $S_{5} \times S_{5} \times S_{5}\times S_{5}$ | 20 | 2.1e-2 | 98.6%
> > > | $C_{11} \times C_{12} \times C_{13} \times C_{14} \times C_{15}$ | 10 | 1.01e-5 | 100%

---

### Author Rebuttal · Authors · 2024-08-07

# NeurIPS Rebuttal

We thank the reviewers for their feedback and insightful comments. We are glad they found our work well-written(__jQuD__, __JC7A__). It is particularly encouraging that many reviewers found our design of architectural constraints to learn group representations novel(__KZHf__), interesting(__JC7A__), and theoretically justified(__jQuD__, __KZHf__). We respond to specific comments below.

__Unclear real-world motivation (JC7A, puxG)
> (__JC7A__) I think the tasks conducted in the experiments are not directly bridged to real problems.
> (__puxG__) The motivation is to solve a mathematical problem or real-world learning task?

The primary motivation of our approach is to assist with mathematical research. The second task used in our experiments over the Artin braid group is a current open research problem. Mathematicians have only found a way to compute the answer for the simplest braid group $B_3$ – and they do not have a simple or intuitive formula. Our goal is to create a model that can help mathematicians build intuition and formulate conjectures by 1) computing additional data points and 2) providing new insights through inspecting the learned representations. We believe this application is relevant to real mathematical research.


## Response to Reviewer jQuD
>It’s unclear why MatrixNet and MatrixNet-MC cannot extrapolate to longer word length.

While MatrixNet and MatrixNet-MC underperform compared to our other variants, it is overstated to say they cannot extrapolate. Despite their high MSE, both approaches maintain relatively high accuracy compared to our baselines. That said, MatrixNet-LN and MatrixNet-tanh have better extrapolation. The reason for this discrepancy is that MatrixNet and MatrixNet-MC have higher relational error indicating they do not learn group representations as accurately. To help make this difference clearer we computed the relational errors of the models on the braid group which we will add to the paper.

| Model | Rel. Error |
| - | - |
| MatrixNet | 14.78 |
| MatrixNet-MC | 5.21 |
| MatrixNet-LN | 0.33 |
| MatrixNet-tanh | 0.45 |

> Should evaluate on groups with different properties.
We chose the symmetric group for our initial experiment in section 5.1 as it is a well-studied group that has the property that all finite groups are isomorphic to a subgroup of a symmetric group. It is also closely connected to the braid group which serves as our motivating problem. We believe that these two groups provide a strong foundation for evaluating MatrixNet but we will perform further evaluations as you suggested.

## Response to Reviewer JC7A
> It is unclear which significantly contributed to the final performance gain: decomposition of g into generators or a trainable representation of a generator.

We test the impact of a trainable representation independent of the decomposition in our experiment in section 5.1 by comparing against the precomputed representation. This ablation replaces the learnable representation with the permutation representation of $S_{10}$ and we see significantly worse performance when compared to learnable representation. We show that just decomposing g into generators but not using a group representation does not result in improved performance by comparing against two sequential baselines, a transformer and LSTM, which take the decomposition as input but do not use a group representation.

> The experiments are not convincing enough—they are relatively small scale, use synthetic tasks only, and have less variety (two tasks).

We disagree that only synthetic tasks are used. The task used in the braid group experiment in section 5.2 is an open math problem from a recent mathematics publication [39]. Mathematicians have only been able to compute the answer for the simplest braid group $B_3$ – and they do not have a simple or intuitive formula. We believe the two groups used, $S_10$ and $B_3$, are sufficiently large with $|S_{10}| = 10!$ and $B_3$ being an infinite group.

## Response to Reviewer KZHf
> It was not clear to me how the learned representations were different from precomputed ones and why they were better for a given task.

Our results parallel other results in deep learning showing that learned feature representations often outperform expert-engineered features. The precomputed representations we used are a natural choice for matrix representations for the groups used. We use the 10x10 permutation matrices to represent $S_{10}$ but this may not be the most useful representation for every task. Our approach is designed to automatically learn a representation that is useful for the given task.

> What happens when the size of the group representations is larger?

Our method can scale to large representations using MatrixNet-MC. MatrixNet-MC assumes a block diagonal structured representation, which is efficient since it means the number of trainable parameters grows asymptotically linearly instead of quadratically. For well chosen block sizes, this does not harm expressivity since many group representations will be block diagonal with respect to a good choice of basis. We include additional MatrixNet results on the braid group task with representation sizes roughly doubled.

| Model | MSE | Acc. |
| - | - | - |
| MatrixNet | 0.975 | 85% |
| MatrixNet-MC | 0.052 | 96% |
| MatrixNet-LN | 4.5e-4 | 100% |
| MatrixNet-tanh | 1.1e-3 | 100% |

## Response to Reviewer puxG

> For the experiment, I think it only solved a mathematical problem at simple setting.

The task used for our braid group experiment in section 5.2 is an open mathematical research problem. This task is limited to $B_3$ since the multiplicity counts are not known for any larger braid groups. The experiment over $S_{10}$ in section 5.1 was chosen to provide a well-studied group to compare precomputed representations against the learned representations of MatrixNet.

---

### Decision · Program_Chairs · 2024-09-25

**Decision:**

Accept (poster)

**Comment:**

The paper proposes to learn symmetry groups so that, in turn, to learn optimal representations for downstream tasks at hand. In particular, MatrixNet is proposed, which takes as input generators, then combines them in a Matrix Block that returns an invertible square matrix. Then for a new group element, its matrix representation is the product of the matrix representations of the respective generators that are needed to generate the group element. A parameterized neural network form of the Matrix Block is proposed, as well as variations. An interesting experiment is on a mathematical problem of predicing Jordan-Holder multiplicities, which to be frank is out of my domain, but seems relevant and interesting for the community.

The reviews are mixed, with most reviewers appreciating the novelty and the presentation of the paper. The main complaint is whether the experiments represent a real-world setting, and if a mathematical problem constitutes really a real-world problem. I think it does, or at least it is an interesting new direction to consider for possible ways to validate machine learning models.

Further, while reading the paper, I think some subsections of related work are missing, specifically on 'Symmetry Discovery', eg

Yang et al., Latent Space Symmetry Discovery, PMLR, 2024
Gabel et al., Learning Lie Group Symmetry Transformations with Neural Networks, TAGML, 2023
Dehmamy et al., Automatic Symmetry Discovery with Lie Algebra Convolutional Networks, NeurIPS, 2021

and in 'Mathematically Constrained Networks' a subsection on mathematical programming for imbuing constraints to neural networks:

Pervez et al., Differentiable Mathematical Programming for Object-Centric Representation Learning, ICLR 2023
Pervez et al., Mechanistic Neural Networks for Scientific Machine Learning, ICML 2024

I suggest the reviewers incorporate the reviewers' suggestions and extend their related work accordingly.